# Realist inquiry into Maternity care @ a Distance (ARM@DA): realist review protocol

Catrin Evans  ,[1] Kerry Evans,[1] Andrew Booth,[2] Stephen Timmons,[3] Nia Jones,[4] Benash Nazmeen,[5] Candice Sunney,[6] Mark Clowes,[2] Georgia Clancy,[1] Helen Spiby[1]

¹School of Health Sciences, University of Nottingham, Nottingham, UK
²ScHARR, The University of Sheffield, Sheffield, UK
³Business School, University of Nottingham, Nottingham, UK
⁴School of Medicine, University of Nottingham, Nottingham, UK
⁵School of Allied Health Professionals and Midwifery, University of Bradford, Bradford, UK
⁶Nottingham Maternity Research Network, Nottingham, UK

**Correspondence to**
Dr Catrin Evans;
catrin.evans@nottingham.ac.uk

## ABSTRACT

**Introduction** One of the most commonly reported COVID-19-related changes to all maternity services has been an increase in the use of digital clinical consultations such as telephone or video calling; however, the ways in which they can be optimally used along maternity care pathways remain unclear. It is imperative that digital service innovations do not further exacerbate (and, ideally, should tackle) existing inequalities in service access and clinical outcomes. Using a realist approach, this project aims to synthesise the evidence around implementation of digital clinical consultations, seeking to illuminate how they can work to support safe, personalised and appropriate maternity care and to clarify when they might be most appropriately used, for whom, when, and in what contexts?

**Methods and analysis** The review will be conducted in four iterative phases, with embedded stakeholder involvement: (1) refining the review focus and generating initial programme theories, (2) exploring and developing the programme theories in light of evidence, (3) testing/ refining the programme theories and (4) constructing actionable recommendations. The review will draw on four sources of evidence: (1) published literature (searching nine bibliographic databases), (2) unpublished (grey) literature, including research, audit, evaluation and policy documents (derived from Google Scholar, website searches and e-thesis databases), (3) expertise contributed by service user and health professional stakeholder groups (n=20–35) and (4) key informant interviews (n=12). Included papers will consist of any study design, in English and from 2010 onwards. The review will follow the Realist and Meta-narrative Evidence Synthesis Evolving Standards quality procedures and reporting guidance.

**Ethics and dissemination** Ethical approval has been obtained from the University of Nottingham, Faculty of Medicine and Health Sciences Ethics Committee (FMHS 426–1221). Informed consent will be obtained for all key informant interviews. Findings will be disseminated in a range of formats relevant to different audiences.

**PROSPERO registration number** CRD42021288702.

## STRENGTHS AND LIMITATIONS OF THIS STUDY

⇒ A realist synthesis moves beyond effectiveness and acceptability to consider evidence related to implementation (what works, for whom and under what contexts).
⇒ Stakeholder engagement (service users and health professionals) is embedded at every stage of the review.
⇒ The focus of a realist review on mechanisms underlying implementation will yield insights that are potentially transferable across geographical and clinical settings.
⇒ Research on digital communication in healthcare is a fast-evolving field, so it is possible that new evidence may be missed, although this may be mitigated to some extent by the iterative approach to searching within a realist review that will support the identification of relevant evidence, even after the initial searches have taken place.

## INTRODUCTION

This article sets out a protocol for a realist review to generate a theory-informed and evidence-informed framework to guide best practice in the implementation of digital clinical consultations in maternity care. The project is being undertaken by a UK-based team and is oriented to producing findings that are actionable in the context of the UK's National Health Service (NHS). Nonetheless, the project will draw on evidence from comparable geographical and health system settings and hence its findings will be of international relevance.

Maternity care in England is undergoing a substantive transformation programme whose overall aim is to support services to work across professional and service boundaries to become safer, kinder, more personalised and more family-friendly.[1–3] A core work stream is 'harnessing digital technology'.[4 5] In the UK, as in many other countries, the COVID-19 pandemic has radically altered and dramatically accelerated the context of this digital policy imperative.[6–9]

One of the most commonly covid-related changes to all forms of service

provision in maternity care has been an increase in the use of remote/virtual consultations using telephone or video.[4 7 10–18] For example, one recent survey of all 194 obstetric units in England (42% response rate) found that 89% reported using digitally delivered consultation methods in antenatal care and 56.8% for postnatal care.[7] For antenatal services, the majority of these (87.7%) were conducted via the telephone, whereas a smaller percentage was conducted using everyday video technology or specialist video technology (12% and 25.9%, respectively). A UK survey of 524 women in 2020 reported that 51.8% had experienced telephone or video consultations.[19]

Within maternity care, consultations have a wide range of potential purposes depending on the individual circumstances and characteristics of the service user, clinical categorisation of the pregnancy (eg, high risk or low risk) and stage of the maternity care pathway (eg, antenatal care, assessment of early labour, postnatal care). Consultations may involve clinical (physical and mental health) assessment, monitoring, safeguarding assessments, health promotion, information giving, education or therapeutic support. The interactions need to be organised and implemented within multiprofessional care pathways that seek to promote continuity, personalisation and choice, recognise diversity and ensure safety.[9] Likewise, consultations need to be supported by auditable records that can be accessed by, and strengthen communications between, different providers and settings.

The rapid shift to remote or digitally delivered clinical consultations was initially implemented to support pandemic response objectives such as social distancing and service demand management.[6 9 16 20–22] Going forward, however, a key challenge for maternity services lies in how to 'repurpose' the use of remote consultations to serve longer term quality, equity and productivity objectives.[6 9] Although remote consultations may now be widespread, the ways in which they can optimally be used in future along maternity care pathways remains unclear. In particular, it is imperative that service innovations do not further exacerbate (and, ideally, should tackle) existing inequalities in service access and clinical outcomes.[9 14 23 24] Hence, there is a need for a comprehensive review of the evidence base to inform future service developments and future research.[6]

### Operational definitions
Digitally-facilitated clinical communication has a highly diverse nomenclature, including terms such as telehealth, telemedicine, virtual/remote care, video consultations and digital consultation (among many others). In spite of efforts by WHO in 2017 to standardise concepts and terminology, there is no internationally agreed or consistent set of terms, meanings or definitions.[25] A similar diversity is found in the rapidly evolving technologies and systems used to implement digitally facilitated consultations. Key distinctions are that digital technologies can be used for: (1) direct 'live' clinical communication/

---

> **Box 1   Operational definition**
>
> *Digital Clinical Consultation (DC-CON)*—Synchronous telephone or video consultations involving direct interaction between a service user and a maternity healthcare professional. It has two-way functionality and can be initiated by either party. It may be linked to, or complemented by, other digital technologies within the maternity care pathways.

consultation between a service user and a practitioner (synchronous) or (2) direct communication that may happen at different time points (asynchronous), such as text messaging. This is in contrast to other digital care modalities such as remote monitoring systems (eg, remote blood pressure monitoring), on-line triage algorithms, the use of apps, wearable personal devices or electronic medical records which are usually asynchronous and do not involve direct interpersonal patient/practitioner communication (although these may be used to trigger, or to directly inform, consultations).

The focus of this project is on maternity care *consultations* that are facilitated through, and/or complemented by, digital technology. To capture this focus, we will use the term 'Digital Clinical Consultation'—henceforth referred to as 'DC-CON'.[26] The working definition for our project is outlined in box 1:

This definition recognises that the focus is on the consultation itself, but that this might be linked to, or informed by, other digital technologies.

This project will investigate digital consultations across the maternity pathway, that is, during pregnancy, the intrapartum period and early postnatal care (up to 10–14 days post-partum).

### Existing evidence on DC-CON in maternity care
There is a large and rapidly expanding literature associated with DC-CON in maternity care.[20] For simplicity, we conceptualise the purpose of DC-CON in two ways (while recognising significant overlap between the two). The first relates to DC-CON that is additional to usual care—where specialist support is required on a single issue where a specific need is identified (eg, interventions to support perinatal mental health, breast feeding or smoking cessation). There is now substantial evidence that such targeted interventions can be feasible, acceptable and effective to varying degrees.[20 27–33]

The second DC-CON purpose relates to situations where DC-CON is already the main modality of care (eg, in telephone helplines or triage systems).[34] It also refers to situations where standard care pathways are altered, so that some in-person points of contact are replaced or supplemented by DC-CON (as happened during the COVID-19 pandemic). DC-CON has been investigated with regards to antenatal care among both high[35 36] and low[37 38]-risk women. For example, a recent randomised control trial in the USA sought to compare standard in-person antenatal care with a new 'hybrid' system that included virtual consultations. This study found higher satisfaction with

care, no difference in health outcomes and less anxiety in the hybrid care group.[37] However, a systematic review of telephone support for women in pregnancy and postpartum was less conclusive, with results suggesting that telephone support may be associated with higher overall satisfaction with care while yielding uncertain impacts on clinical outcomes.[30]

Overall, existing research on DC-CON in maternity care shows that it can be safe and acceptable in controlled conditions.[27–29 35 37–41] Research also shows that the experiences of staff and women with DC-CON vary quite significantly.[7 12 34 39 42–50] A gap remains, therefore, in understanding the conditions required for safe and acceptable DC-CON implementation and in understanding the factors that underpin variation in experience. For example: variations between women of different ethnicities (to ensure that DC-CON does not further exacerbate racial inequalities in maternity care)[51]; how women who face communication barriers will be supported by workers to understand their care and make choices via DC-CON[16 52–55]; and how access to the technologies necessary for DC-CON will vary between service users and service providers as a result of financial resources, internet connectivity, digital literacy and the digital maturity of NHS services.[4 52 56] Such variation is problematic, as it is implementation in real-world contexts, at scale, that needs to be understood for the systemwide transformation as envisaged in the NHS Maternity Transformation Programme.[2 21 57] It is this gap that the proposed project will address.

### Research aim and question

This project aims to undertake an in-depth and theory-informed investigation of the evidence around implementation of digital clinical consultations. It seeks to illuminate how digital consultations can work to support safe, personalised and appropriate maternity care and to clarify when they might be most appropriately used, for whom, when and in what contexts.

Scoping searches of PROSPERO, Medline and the Cochrane Library have not identified any similar reviews currently being undertaken.

### Philosophy and methodology

We will undertake a realist synthesis in which patient and public involvement (PPI) and diverse stakeholder participation are embedded at every stage.[58] Realist syntheses seek to investigate the relationships between what can be observed and experienced, human interpretation of this reality and 'unseen' underlying social structures.[59–61] Increasingly, realist approaches are applied in health research to investigate the complexity of health-related behaviour and to explore how behaviours are shaped by human agency via intersecting social structures that manifest differently in different contexts.[62] Realist approaches are particularly well suited to investigating complex interventions such as DC-CON.[63 64] This is because realist inquiry seeks to establish causal relationships—expressed as 'programme theories'—between intersecting intervention components, contexts and outcomes.[65] When applied to evidence synthesis, a realist approach focuses on understanding *how* particular interventions lead to particular outcomes under particular contexts (ie, 'how does A lead to B'? and 'how might B be affected when A is implemented through contexts C, D or E'?).[66] This contrasts with the linear or deterministic approaches adopted by conventional systematic reviews which ask: 'does A lead to B'? (eg, an effectiveness review) or: 'is A acceptable or meaningful to a particular group of individuals in a particular context'? (eg, a qualitative review).[58]

As applied to our research question, the logic of a realist review proposes that different types of interventions (eg, a video call or a phone call) supply particular resources into a situation that prompt diverse possible reactions and responses (also referred to as 'reasonings') from women and health professionals.[67] The interaction between the resource and the response constitutes a 'mechanism'.[67] Mechanisms are influenced not only by differences in how an intervention is delivered but also by the context of particular women's lives or different service configurations.[67] This means that the acceptability and outcomes of DC-CON may be highly contextually contingent.[68] A realist approach seeks to identify certain patterns ('demi-regularities'[59] between types of resource (eg, a phone call)), how individuals respond and how a particular context may alter these responses. Hence, a realist review is focused on identifying and testing programme theories that can account for the contingent nature of intervention implementation.[69] These mid-range explanatory theories are a principal output from a realist review. Their insights make a substantive contribution to health service innovation because, by taking context into account, they are potentially 'transferable'—producing insights relevant to a wide range of settings and geographical contexts.[70]

Our proposed realist synthesis will be conducted in four iterative phases (see below). The review will follow the Realist and Meta-narrative Evidence Synthesis Evolving Standards (RAMESES) quality procedures[69] and, subsequently, will comply with RAMESES reporting guidance.[71]

Each phase of the review potentially draws on three sources of evidence: (1) literature (published and unpublished research, audit, evaluation and theory), (2) diverse stakeholder expertise and insights and (3) key informant interviews.

### METHODS AND ANALYSIS
### Patient and public involvement

This project has been informed by service user involvement from its outset and seeks to follow the National Institute for Health Research (NIHR) UK Standards for Public Involvement in Research.[72] Public involvement in the project will be reported following the Guidance for Reporting Involvement of Patients and the Public (2) (GRIPP2) reporting checklist.[73]

A maternity research priority setting exercise was undertaken in 2020 by some members of the research team in which a PPI research network played a key role (the Nottingham Maternity Research Network - NMRN).[74] DC-CON was identified as an important topic for future work. A member of the NMRN (CS) subsequently became a coinvestigator on the current project. The NMRN has been instrumental in shaping the current project. For example, they identified a need to increase the diversity of PPI within the project. As a result, two other organisations were invited to become involved: (1) Women, Health and Family Services (based in east London serving deprived communities in an ethnically and linguistically diverse area and which runs an award winning 'Maternity Mates' community volunteer service) and (2) the National Autistic Society (this has national representation and has been undertaking several recent projects on neurodiversity and pregnancy/motherhood). Within the project structure, these two organisations have been grouped together with the NMRN to form a 'Community Organisation and Service User Stakeholder Group' (COSU-SG). The COSU-SG enables a wide spectrum of socioeconomic, geographical, ethnic and neurodiversity among women to be represented. Each of the three organisations has a named lead for involvement in the project. Up to five individuals from each organisation will attend project stakeholder events (the organisational lead and four others). The organisational leads are responsible for selecting the most appropriate individuals from their organisations to engage with project events and will provide them with ongoing support and information. They will also facilitate consultation and engagement with their wider networks where required (eg, through social media or email lists or by suggesting particular individuals as key informants).

In addition to the COSU-SG, all project phases will also be informed by a health professional stakeholder group (HP-SG). The HP-SG will include between 10 and 20 obstetricians, digital midwives, frontline and managerial midwifery staff from different maternity settings—recruited via social media and through professional networks. Group membership is not bounded and fixed, however, but may evolve as the project progresses to ensure that appropriate expertise and insights are accessed. The purpose of the two stakeholder/PPI groups is to shape, participate in, contribute to and advise the research team on all project phases to ensure that project questions, programme theories and recommendations reflect diverse real-world experiences and are relevant.

For both groups of stakeholders, anyone expressing an interest to get involved will first be contacted by the project PI (CE) for an in-depth discussion about the project and their potential contribution and commitment. They will also be sent written information about the study (this information sheet will be reviewed for readability/plain English). It will be stressed that their involvement is entirely voluntary and that they can withdraw at any time. Subsequently, the PPI coapplicant (CS) will contact individuals periodically to offer an opportunity for further discussion, to ask questions and to ensure that they remain happy to contribute. All personal details about stakeholders and records of project meetings/workshops will be securely stored in password-protected files according to a University-approved Data Management Plan.

## Overarching approach to literature searching

Within this realist review, the overall search approach will follow the only published systematic approach to the 'realist search'.[75 76] This approach extends and enhances the Task and Time Template for a realist review advanced by Pawson.[59] It outlines four separate and distinct phases of searching using different retrieval techniques and targeted at different evidence bases, 'topped' and 'tailed' by precise question formulation and meticulous documentation.[75]

The search strategies within each phase will be developed and operationalised by experienced information specialists (AB and MC). The search dates will be restricted to evidence from 2010 to the present to reflect the need for contemporary data. Likewise, inclusion of evidence will be limited to reports and studies in the English language and in high-income settings (Organisation for Economic Co-operation and Development (OECD) countries). The search approach will engage the project advisory group and stakeholders throughout, consulting them where required to provide advice (eg, clarifying the terminology used at service level) and help with identifying relevant literature (including, eg, cascading requests to identify literature and policy/practice documents through their associated social media sites and websites).

## Key informant interviews

Data sources for the project may also involve a small number of key informant interviews. Within a realist review, the purpose of these is to provide additional insights to confirm, refute or refine proposed programme theories.[58] These are most likely to be part of phase 3 (see below) but may also take place in the other phases if additional data are required to support programme theory development. The specific function of the interviews within a realist review means that only a small number are usually required—we anticipate no more than 12 (although the exact number is flexible and will depend on the nature and extent of gaps in the evidence). Potential interviewees may include health professionals or service users. They will be recruited via calls put out through social media and professional networks with the aim of providing new insights or testing the transferability of programme theories. Those expressing an interest will be followed up by phone/online call and provided with further information about project. Those who agree to proceed will be required to complete an online consent form. Interviews will be undertaken remotely, audio-recorded and analysed using a framework approach,[77] mapping the data to the Context-Mechanism-Outcome

(CIMO) configurations. Service users agreeing to an interview will be provided with a £25 Amazon voucher to recognise their time contribution.

### Review phases

Each phase of the review (with its associated search approaches) is described sequentially below, although in practice, there is considerable iteration between them (see online supplemental file 1 for a flowchart of the proposed process).

### Phase 1: refining the review scope and developing initial programme theories

This phase aims to identify and make explicit (through CMO configurations) an initial set of programme theories that may explain how DC-CON in maternity care can be used to achieve optimal outcomes, for whom and in what contexts? Given the potential variability of modality and use of DC-CON in maternity care, an important part of this process is to focus and prioritise the most important questions and outcomes.

Phase 1 involves undertaking some broad background searches to assess the breadth, depth and range of evidence available. The team will then seek to identify key papers (of any study design) on DC-CON implementation that can yield insights into programme theory development. The search is iterative, utilising searches on electronic databases, grey literature sources, suggestions from stakeholders, citation tracking and reference list searching of conceptually rich index papers. This stage will include, but will not be restricted to, maternity care settings as we know from pr-protocol scoping that relevant theories have been developed regarding the implementation of DC-CON in other clinical settings.[26 78–80] Each included paper will be scrutinised to elucidate how 'best practice' in DC-CON is defined and to identify the mechanisms through which successful consultations are purported to work in relation to different contextual configurations and population groups. Details of key theories ('candidate theories') that have been used to explain implementation mechanisms will be extracted. Excel will be used to extract the key characteristics of each paper and NVivo will be used to code the key findings related to implementation. Where possible, findings will be coded and analysed in relation to possible context, mechanism and outcome configurations, helping to generate initial ideas around relevant programme theories. The coding and analysis templates for the reviews in this stage will be developed and piloted by several members of the review team. The majority of the coding will be undertaken by one reviewer, with a second reviewer completing a random sample of approximately 20% to monitor quality and consistency. The findings of this stage will be presented using tables, figures, flowcharts and narrative summaries to highlight the key features of the evidence and to describe potential programme theories.

Phase 1 includes two stakeholder workshops (at the beginning and end) to help generate initial programme theories and to refine the key focus of the review.

### Phase 2: evidence retrieval, review and synthesis

The aim of phase 2 is to determine whether the initial programme theories are supported by empirical evidence and to analyse this evidence to elaborate, refine, adjust and test the theories. This phase continues the iterative process of literature searching, data extraction and analysis. A core focus is to search for empirical studies that can provide data with which to explore and elaborate the initial programme theories. The search strategy will be iterative, proceeding with carefully formulated searches based on subsets of literature constructed using terms associated with the initial programme theories (CIMO frameworks) and key concepts.[75 81] Searches are initially broad but may then be narrowed down to focus more specifically on evidence associated with particular mechanisms. As an example, in our review, we may construct an initial search using sets of terms derived from our programme theories combined using the 'AND' Boolean operator:

► Context (from the initial programme theory)—postnatal care.
► Interventions/phenomenon of interest (from the initial programme theory)—video-based consultation.

A follow-up search might then focus more specifically on the influence of a mechanism (eg, trust) in relation to an outcome (eg, positive health behaviours). In the first instance, the main focus of the search for evidence will be for literature related directly to maternity settings. However, in a realist review, the focus of analysis is the programme theory (or mechanism of action)—hence, we may also draw on wider literature to seek opportunities for transferable learning. In the example above, we might, therefore, seek evidence related to 'trust' in the context of video consultations from other clinical settings such as primary care to confirm or refute our emerging theories.

The searches will include systematic reviews and empirical research of any study design, including service evaluation, audit and quality improvement projects. The searches will also include existing policies and practice guidelines surrounding DC-CON in maternity care in the UK. This is because the recommendations set out in policy or practice guidance rest on implicit or explicit theoretical assumptions regarding implementation. Moreover, the inclusion of policy documents will help to ensure that governance issues are considered alongside implementation, so that the review is appropriately contextualised.

As in phase 1, search sources will include electronic databases, grey literature and expert stakeholders (see table 1 for sources of evidence and online supplemental file 2) for an initial search strategy for OVID MEDLINE—which will be adapted for different databases). Additional search approaches will include reference list searching, citation tracking, identification of sibling papers (linked papers from a single study) as well as cluster searching,[81] which involves building up rich 'cases' of different models of DC-CON in order to grow a cluster of related reports around named or identifiable initiatives to offer

| Table 1 | Sources of evidence | |
|---|---|---|
| **Electronic databases (2010–present)** | **Grey literature sources** | **Stakeholders** |
| ► Cochrane Central Register of Controlled Trials<br>► Cochrane Database of Systematic Reviews<br>► JBI Library<br>► MEDLINE Ovid<br>► Embase Ovid<br>► PsycINFO Ovid<br>► ASSIA Cambridge Scientific Abstracts (Applied Social Sciences Index and Abstracts)<br>► CINAHL EBCSCOhost (Cumulative Index to Nursing and Allied Health Literature)<br>► MIDIRS Ovid | ► Google Scholar<br>► Websites (eg, RCOG, RCM, RCN, NCT, NHS Trusts, NHSX, Health Foundation, WHO)<br>► Conference proceedings<br>► OpenGrey<br>► ProQuest Dissertations & Theses<br>► EThOS – British Library Electronic Theses Online | ► Project Advisory Group<br>► Stakeholder groups<br>► Others (eg, via social media requests, email list serves) |

both richness and detail. Unlike a conventional systematic review search, searches in a realist review are not necessarily exhaustive but follow the principles of theoretical saturation, ceasing when programme theories are deemed to be sufficiently explained, supported or refuted by the empirical evidence.[69 75] Likewise, additional targeted searches may be undertaken to explore new mechanisms or other aspects of programme theory that may be identified during the review.[69 75]

Records will be imported into EndNote and duplicates were removed. Study screening and selection will be undertaken by two reviewers independently, with recourse to other team members in cases of ambiguity or disagreement. To aid transparency and consistency, once the initial programme theories have been formulated, an 'inclusion criteria flowchart' will be constructed in which key concepts are operationalised and against which each potential study can be assessed.[82]

Records will initially be screened by title and abstract. All seemingly relevant papers will then be examined in full text and reasons for exclusion noted in a table. In line with realist methodology, records will be screened for inclusion based on relevance, rigour and richness.[58 69] An assessment of relevance considers the extent to which a paper can directly contribute to theory building or theory testing.[69] An assessment of rigour considers whether the methods used to generate the relevant data are credible and trustworthy. Richness refers to the extent to which study findings are fully elaborated through 'thick description', grounded in the data and provide explanatory insights.[83] In a realist review, the process of quality assessment refers to the specific data that is relevant to the particular programme theory being examined rather than on the basis of a global evaluation of overall study quality.[84] Hence, for each key finding, a judgement needs to be made about the plausibility and coherence of the methods used to generate it.[71] For each included paper, the team will follow the process outlined by Rycroft-Malone *et al*,[66] asking: *is the evidence provided in this theory area good enough and relevant enough to be included?*. These judgements will be articulated and recorded for each study as part of the screening and data extraction process.

Data will be extracted from the included studies in two ways. First, information about study characteristics will be extracted into a summary table (as is the case with a conventional systematic review). This will include information on features such as study setting, design, methods, details of intervention modality and technology, participants, outcomes and stage of maternity care pathway. Second, a bespoke data extraction form will be developed based on the initial programme theories, and, as noted above, will include sections in which to note assessments of relevance, richness and rigour. Theory-based data extraction enables the coding and charting of relevant data, so that elements of the theory can become populated to investigate what works, for whom, how and in what contexts. The analytical process involves both deductive and inductive coding. Deductive coding will involve extracting data that appears to be directly related to aspects of the programme theory. Where it is possible to make relevant inferences, the data will also be coded in relation to contexts, mechanisms or outcomes. However, the evidence may also reveal new contexts, mechanisms or outcomes, which will be identified and coded inductively.

The data extraction templates and associated analytical process will be developed collaboratively among the core research team and piloted extensively. Once the team feel they have achieved a clear and consistent approach, the remaining data extraction will be undertaken by one reviewer, with a second reviewer checking approximately 20%. The outputs of this stage will be a set of evidence tables. There will be one overarching table representing the key characteristics of all the studies included in the review. There will then be a series of tables organised to represent the different bodies of literature and findings related to each initial programme theory. Thus, each theory area will have its own evidence table.

The process of data analysis will be ongoing and iterative. The evidence will be reviewed within and across the theory areas to explore how it builds on, refutes or provides alternative explanations for the initial CMO configurations. The analytical process will involve asking questions such as: *What does this evidence suggest about this aspect of our theory? Does it support it? Does it disprove it?*

> **Box 2    Analytic process of a realist review**
>
> ⇒ Juxtaposition of sources of evidence—for example, where evidence about outcomes in one study allows insights into evidence about outcomes in another study.
> ⇒ Reconciling of sources of evidence—where results differ in comparable circumstances, these will be examined further to find possible reasons for the different results.
> ⇒ Adjudication of sources of evidence—based on methodological strengths or weaknesses.
> ⇒ Consolidation of sources of evidence—where outcomes differ in particular contexts, an explanation will be constructed on how and why these outcomes occur differently.
> ⇒ Situating sources of evidence—when outcomes are different in particular contexts, a possible explanation will be developed as to why they differ.

*Does it suggest an amendment to it?*[69] This analytic process involves both abductive and retroductive reasoning—that is, making new observations from the evidence, inferring plausible explanations related to the programme theory, seeking to understand the cause of perceived events beyond what can be observed and seeking to identify overarching patterns. Wong *et al*[69] propose a series of 'conceptual tools' (derived from Pawson,[59] which will be employed by the review team in this complex process, as indicated in box 2 below.

Our stakeholder groups will continue to be consulted at different points to seek their views and test out new ideas.

An important feature of the analysis will be to explore the issue of temporality and sustainability within the evidence. For example, it might be expected that evidence on COVID-19-related DC-CON implementation might include different experiences depending on the stage of the pandemic and the length of time that staff and service users have had to adjust to changes in service delivery models. Likewise, even in non-COVID-related studies, experiences with new technologies vary depending on the length of time since introduction, with many studies focusing on the early stages of implementation rather than exploring how and why technologies become embedded, normalised or sustainable.[80 85]

In this project, we will endeavour to take temporality into account in two ways. The first, relates to evidence that is specifically concerned with changes made during the COVID-19 pandemic. Here, we will ensure that our data extraction template explicitly acknowledges the time period of data collection, extracting and mapping data to the stages of the pandemic and then transparently considering these stages as part of the analysis. Likewise, working with our advisory and stakeholder groups, we will attempt to develop a clearer picture of how digital consultations were initially introduced in different settings and how their use and implementation may have adapted and changed as the pandemic progressed. The second issue relates to the fact that not all of the evidence that we anticipate including in the review will come from the pandemic

context. We will endeavour to include temporality as an aspect of our analysis for all the evidence by explicitly considering the stage of implementation of an innovation that is reported and by considering data related to change and adaptation over time (where reported). We plan to draw on concepts from the Dynamic Sustainability Framework[86] to help 'sensitise' the team to issues of temporality. The framework focuses attention to the constantly changing evidence-base, multilevel context in which interventions are delivered, and the broader ecological system within which the maternity care exists and operates.

The output of phase 2 will be a comprehensive set of evidence tables and refined programme theories linked to associated sets of working papers organised by DC-CON type and purpose in relation to different points in the maternity care pathway.

### Phase 3: Test and refine programme theories (validation)
The purpose of this phase is to test and further refine the programme theories. This is done in three interlinked ways.

First, as part of an ongoing recursive approach, new literature searches will be undertaken to find, test and empirically explore any new theories identified during the review process.[75] This may also include a need to revisit and recode previously identified papers. As noted by Booth *et al*,[75] this stage may involve searches for specific named theories that have been identified in Phase 2 and that are considered worthy of further empirical exploration.

Second, further in-depth consultation/workshops with stakeholders will be undertaken to test and refine proposed theories. Our stakeholder groups will be asked to consider the proposed theories in relation to their own experiences, paying particular attention to clarifying the key mechanisms that are required to produce desired outcomes and to identify the key links between these mechanisms and different contexts. This process will help to identify 'gaps' that remain in our understanding and to clarify the essential elements of intervention and context that need to be placed to ensure that the appropriate mechanisms can be activated.

Third, as described above, we will undertake a limited number of key informant interviews to test aspects of the programme theories and to evaluate their plausibility, credibility and transferability.[58]

The output of this phase will be a theoretically-grounded explanatory framework for safe, appropriate and acceptable DC-CON in maternity care that can be used to guide intervention development, policy and practice.

### Phase 4: Development of actionable recommendations
The purpose of this phase is to generate actionable recommendations from the review. We will do this by holding an online webinar to which a much larger group of stakeholders will be invited. The purpose of this webinar will be to share and 'sense-check' the review

findings and to solicit suggestions for recommendations, actions and strategies for dissemination. In addition, we will hold further meetings to discuss, develop and finalise recommendations appropriate to a range of constituents (researchers, commissioners, service leads, professionals, women, community organisations) and feasible for implementation in a UK NHS context.

## ETHICS AND DISSEMINATION

Ethical approval for the key informant interviews has been obtained from the University of Nottingham Faculty of Medicine and Health Sciences Research Ethics Committee—reference no. FMHS 426–1221. Formal ethical approval for the stakeholder involvement/consultations is not required; however, the principles of research integrity and informed consent will be followed at all times.

Dissemination strategies will be informed through ongoing engagement with our stakeholder groups. Key strategies will include academic papers, conference presentations, development of an online educational resource for health professionals and short papers in 'practice-focused' journals.

**Contributors** The original protocol was developed with extensive conceptualisation and written input from all authors. CE prepared the protocol as a manuscript for publication. All authors have read, commented on and approved the final manuscript. AB contributed inputs on realist methodology. MC provided inputs on the search strategy. ST provided inputs on implementation science issues. KE, HS, BN and NJ contributed sections on maternity policy and practice. CS provided guidance on conceptualising issues related to PPI and stakeholder involvement. GC contributed to the revisions required following peer review and drafted the response to reviewers.

**Funding** This work is supported by the UK National Institute for Health Research (NIHR), Health Services Delivery Research Programme, grant number: NIHR134535.

**Competing interests** None declared.

**Patient and public involvement** Patients and/or the public were involved in the design, or conduct, or reporting, or dissemination plans of this research. Refer to the Methods section for further details.

**Patient consent for publication** Not applicable.

**Provenance and peer review** Not commissioned; externally peer reviewed.

**Data availability statement** As this is a protocol, no data are currently available.

**ORCID iD**
Catrin Evans http://orcid.org/0000-0002-5338-2191

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
