## [Reviewer comments · BMJ Open]

ARTICLE DETAILS

TITLE (PROVISIONAL)	A Realist Inquiry into Maternity Care @ a Distance (ARM@DA): Realist Review Protocol
AUTHORS	Evans, Catrin; Evans, Kerry; Booth, Andrew; Timmons, Stephen; Jones, Nia; Nazmeen, Benash; Sunney, Candice; Clowes, Mark; Clancy, Georgia; Spiby, Helen

VERSION 1 – REVIEW

REVIEWER	Rayment-Jones, Hannah Kings Coll London, Women and Childrens Health
REVIEW RETURNED	28-Mar-2022

GENERAL COMMENTS	A clear, detailed protocol for a realist inquiry into digital clinical consultations in maternity care. The gaps in knowledge could be more clearly defined, for example further discussion around the variation in outcomes and experiences mentioned in the introduction. The methods proposed appear relevant to the research question and include a high standard of service user engagement.
---

VERSION 1 – AUTHOR RESPONSE

Reviewer	
The gaps in knowledge could be more clearly defined, for example further discussion around the variation in outcomes and experiences mentioned in the introduction	Thank you for this comment. We have added in further detail into the section entitled ' Existing evidence on DC-CON ' (p.6) – aiming to highlight and outline specific inequalities and other reasons for potential variation in DC-CON outcomes.